# DIRECT OPTIMIZATION THROUGH $\arg\max$ FOR DISCRETE VARIATIONAL AUTO-ENCODER

## ABSTRACT

Reparameterization of variational auto-encoders is an effective method for reducing the variance of their gradient estimates. However, when the latent variables are discrete, a reparameterization is problematic due to discontinuities in the discrete space. In this work, we extend the direct loss minimization technique to discrete variational auto-encoders. We first reparameterize a discrete random variable using the $\arg\max$ function of the Gumbel-Max perturbation model. We then use direct optimization to propagate gradients through the non-differentiable $\arg\max$ using two perturbed $\arg\max$ operations.

## 1 INTRODUCTION

Models with discrete latent variables drive extensive research in machine learning applications, such as language classification and generation (Yogatama et al., 2016; Hu et al., 2017; Shen et al., 2018), molecular synthesis (Kusner et al., 2017), or game solving (Mena et al., 2018). Compared to their continuous counterparts, discrete latent variable models can decrease the computational complexity of inference calculations, for instance, by discarding alternatives in hard attention models (Lawson et al., 2017), they can improve interpretability by illustrating which terms contributed to the solution (Mordatch & Abbeel, 2017; Yogatama et al., 2016), and they can facilitate the encoding of inductive biases in the learning process, such as images consisting of a small number of objects (Eslami et al., 2016) or tasks requiring intermediate alignments (Mena et al., 2018). Finally, in some cases, discrete latent variables are natural choices, including when modeling datasets with discrete classes (Rolfe, 2016; Jang et al., 2016; Maddison et al., 2016).

Models involving discrete latent variables are hard to train, with the key issue being to estimate the gradients of the resulting non-differentiable objectives. While one can use unbiased estimators, such as REINFORCE (Williams, 1992), their variance is typically high (Paisley et al., 2012; Mnih & Gregor, 2014; Titsias, 2015; Gu et al., 2015; Mnih & Rezende, 2016; Tucker et al., 2017). In variational auto-encoders (VAEs) with continuous latent variables, the reparameterization trick provides a successful alternative (Kingma & Welling, 2013; Rezende et al., 2014). However, it cannot be directly applied to non-differentiable objectives.

Recent work (Maddison et al., 2016; Jang et al., 2016) uses a relaxation of the discrete VAE objective, where latent variables follow a Gumbel-Softmax distribution. The Gumbel-Softmax approach is motivated by the connection of the Gumbel-Max perturbation model and the Gibbs distribution (Papandreou & Yuille, 2011; Tarlow et al., 2012; Hazan et al., 2013; Maddison et al., 2014). The Gumbel-Softmax distribution relaxes the non-differentiable Gumbel-Max trick for sampling from the categorical distribution by replacing the $\arg\max$ operation with a softmax operation. The reformulation results in a continuous objective function, which allows the use of the reparameterization trick. (Mena et al., 2018) provide an extension of this approach for discrete latent structures, namely distributions over latent matchings.

Our work proposes minimizing the non-differentiable objective, by extending the direct loss minimization technique to generative models (McAllester et al., 2010; Song et al., 2016). Since categorical variables are represented by the Gibbs distribution, we start from the $\arg\max$ formulation of the Gibbs distribution. We subsequently derive an optimization method that can propagate (biased) gradients through reparameterized $\arg\max$. The gradient of the resulting expectation is estimated by the difference of gradients of two $\arg\max$-perturbations.

We begin by introducing the notation, the VAE formulation and the equivalence between the Gibbs distribution and the Gumbel-Max perturbation model in Section 3. In Section 4.1, we use these components to reformulate the discrete VAE objective, with respect to the $\arg\max$ prediction operation. We subsequently state and prove the main result that allows us to differentiate through the $\arg\max$ function in Section 4.2. In Section 5, we extend this result to mixed discrete-continuous VAEs and to semi-supervised VAE objectives. Finally, we demonstrate the effectiveness of our approach on image generation.

## 2 RELATED WORK

Variational inference has been extensively studied in machine learning, see Blei et al. (2017) for a review paper. In our work, we consider variational Bayes bounds with a discrete latent space. Many approaches to optimizing the variational Bayes objective, that are based on samples from the distribution, can be seen as applications of the REINFORCE gradient estimator (Williams, 1992). These estimators are unbiased, but without a carefully chosen baseline, their variance tends to be too high for the estimator to be useful and considerable work has gone into finding effective baselines (Paisley et al., 2012). Other methods use various techniques to reduce the estimator variance (Ranganath et al., 2014; Mnih & Gregor, 2014; Gu et al., 2015).

Reparameterization is an effective method to reduce the gradient estimate variance in generative learning. Kingma & Welling (2013) have shown its effectiveness in auto-encoding variational Bayes (also called variational auto-encoders, VAEs) for continuous latent spaces. Rezende et al. (2014) demonstrated its effectiveness in deep latent models. The success of these works led to reparameterization approaches in discrete latent spaces. Rolfe (2016) and Vahdat et al. (2018) represent the marginal distribution of each binary latent variable as a continuous variable in the unit interval. This reparameterization allows the backpropogation of gradients through the continuous representation. These works are restricted to binary random variables, and as a by-product, it encourages high-dimensional representations for which inference is exponential in the dimension size. In contrast, our work reparameterizes the discrete Gibbs latent model, using a Gumbel-Max perturbation model and directly propagates gradients through the reparameterized objective.

Maddison et al. (2016) and Jang et al. (2016) recently introduced a novel distribution, the Concrete distribution or the Gumbel-Softmax, that continuously relaxes discrete random variables. Replacing every discrete random variable in a model with a Concrete random variable, results in a continuous model, where the reparameterization trick is applicable. These works are close to ours, with a few notable differences. They use the Gumbel-Softmax function and their model is smooth and reparameterization may use the chain rule to propagate gradients. Similar to our setting, the Gumbel-Softmax operation results in a biased estimate of the gradient. Different from our setting, the softmax operation relaxes the variational Bayes objective and results in a non-tight representation. Our work uses the Gumbel-Max perturbation model, which is an equivalent representation of the Gibbs distribution. With that, we do not relax the variational Bayes objective, while our $\arg\max$ prediction remains non-differentiable and we cannot naively use the chain rule to propagate gradients. Instead, we develop a direct optimization method to propagate gradients through $\arg\max$ operation using the difference of gradients of two max-perturbations. Our gradient estimate is biased, except for its limit.

Differentiating through $\arg\max$ prediction was previously done in discriminative learning, in the context of direct loss minimization (McAllester et al., 2010; Song et al., 2016). Unfortunately, direct loss minimization cannot be applied to generative learning, since it does not have a posterior distribution around its $\arg\max$ prediction. We apply the Gumbel-Max perturbation model to transform the $\arg\max$ prediction to the Gibbs distribution. This also allows us to overcome the "general position" assumption in (McAllester et al., 2010; Song et al., 2016) using our "prediction generating function".

## 3 BACKGROUND

To model the data generating distribution, we consider samples $S = \{x_1, ..., x_m\}$ originating from some unknown underlying distribution. We explain the generation process of a parameterized model $p_\theta(x)$, by minimizing its log-loss when marginalizing over its hidden variables $z$. Using variational

Bayes, we upper bound the log-loss of an observed data point

$$\sum_{x \in S} -\log p_\theta(x) \le \sum_{x \in S} -\mathbb{E}_{z \sim q_\phi} \log p_\theta(x|z) + \sum_{x \in S} KL(q_\phi(z|x)||p_\theta(z)) \tag{1}$$

Typically, the model distribution $p_\theta(x|z)$ follows the Gibbs distribution law $p_\theta(x|z) = e^{-\theta(x,z)}$. When considering a discrete latent space, i.e., $z \in \{1, ..., k\}$, the approximated posterior distribution follows the Gibbs distribution law $q_\phi(z|x) \propto e^{\phi(x,z)}$. The challenge in generative learning is to reparameterize and optimize

$$-\mathbb{E}_{z \sim q_\phi} \log p_\theta(x|z) = \sum_{z=1}^{k} \frac{e^{\phi(x,z)}}{\sum_{\hat{z}} e^{\phi(x,\hat{z})}} \theta(x, z) \tag{2}$$

In our work, we reparameterize the variational bound using the equivalence between Gibbs models and Gumbel-Max perturbation models.

Gumbel-Max perturbation models allow an alternative representation of Gibbs distributions $q_\phi(z|x) \propto e^{\phi(x,z)}$ that is based on the extreme value statistics of Gumbel-distributed random variables. Let $\gamma$ be a random function that associates random variables $\gamma(z)$ for each $z = 1, ..., k$. When the random perturbations follow the zero mean Gumbel distribution law, whose probability density function is $g(\gamma) = \prod_{z=1}^{k} e^{-(\gamma(z)+c+e^{-(\gamma(z)+c)})}$ for the Euler constant $c \approx 0.57$, we obtain the following identity between Gibbs models and Gumbel-Max perturbation models[1] (cf. Kotz & Nadarajah (2000))

$$\frac{e^{\phi(x,z)}}{\sum_{\hat{z}} e^{\phi(x,\hat{z})}} = \mathbb{P}_{\gamma \sim g}[z^{\phi+\gamma} = z], \text{ where } z^{\phi+\gamma} \stackrel{def}{=} \arg \max_{\hat{z}=1,...,k} \{\phi(x, \hat{z}) + \gamma(\hat{z})\} \tag{3}$$

For completeness, a proof for this statement appears in Appendix A.

## 4 REPARAMETERIZATION AND DIRECT OPTIMIZATION

In the following section, we reformulate the discrete VAE objective using the Gumbel-Max perturbation model. We then derive an optimization method that directly propagates gradients through the reparameterized $\arg \max$ function.

### 4.1 VAEs WITH GUMBEL-MAX PERTURBATION MODELS

Perturbation models allow an alternative representation of Gibbs distributions $q_\phi(z|x) \propto e^{\phi(x,z)}$. Using the Gumbel-Max perturbation model in Equation (3), the negative log likelihood in Equation (2) takes the form

$$-\mathbb{E}_{z \sim q_\phi} \log p_\theta(x|z) = \sum_{z=1}^{k} \mathbb{P}_{\gamma \sim g}[z^{\phi+\gamma} = z]\theta(x, z) = \sum_{z=1}^{k} \mathbb{P}_{\gamma \sim g}[z^{\phi+\gamma} = z]\theta(x, z^{\phi+\gamma}) \tag{4}$$

$$= \sum_{z=1}^{k} \mathbb{E}_{\gamma \sim g}[\mathbf{1}_{z^{\phi+\gamma}=z}\theta(x, z^{\phi+\gamma})] = \mathbb{E}_{\gamma \sim g}[\theta(x, z^{\phi+\gamma})] \tag{5}$$

These quantities result from applying the law of total expectation, while realizing the probability events for $z^{\phi+\gamma}$: The last equality in Equation (4) holds since we restrict to the space $z^{\phi+\gamma} = z$. The last equality in Equation (5) holds by the linearity of the expectation, i.e., $\sum_{z=1}^{k} \mathbb{E}_{\gamma \sim g}[\mathbf{1}_{z^{\phi+\gamma}=z}\theta(x, z^{\phi+\gamma})] = \mathbb{E}_{\gamma \sim g}[\sum_{z=1}^{k} \mathbf{1}_{z^{\phi+\gamma}=z}\theta(x, z^{\phi+\gamma})]$ and the fact that $\sum_{z=1}^{k} \mathbf{1}_{z^{\phi+\gamma}=z} = 1$.

The gradient of the decoder log probability $\theta(x, z^{\phi+\gamma})$ with respect to its parameters is derived by the chain rule. The main challenge is to evaluate the gradient of $\mathbb{E}_{\gamma \sim g}[\theta(x, z^{\phi+\gamma})]$ with respect to the encoder parameters, since the chain rule does not propagate through the $\arg \max$ function $z^{\phi+\gamma}$.

---

[1]The set $\arg \max_{\hat{z}=1,...,k}\{\phi(x, \hat{z}) + \gamma(\hat{z})\}$ is the set of all maximal arguments, and does not always consist of a single element. However, since the Gumbel distribution is continuous, the $\gamma$ for which their set $\arg \max_{\hat{z}=1,...,k}\{\phi(x, \hat{z}) + \gamma(\hat{z})\}$ consists more than a single element has a measure of zero. For notational convenience, when we consider integrals (or probability distributions), we ignore measure zero sets.

### 4.2 DIRECT OPTIMIZATION THROUGH $\arg\max$

Our main result is presented in Theorem 1 and shows how to compute the gradient of the reparameterized discrete VAE, i.e., $\mathbb{E}_{\gamma \sim g}[\theta(x, z^{\phi+\gamma})]$ with respect to the encoder parameters $v$. In the following we omit $\gamma \sim g$ for brevity. To make the encoder parameters $v$ explicit in our notation, we denote the encoder function by $\phi_v(x, z)$.

Instrumental to our approach, is a novel "prediction generating function".

$$G(v, \epsilon) = E_\gamma[\max_{\hat{z}}\{\epsilon\theta(x, \hat{z}) + \phi_v(x, \hat{z}) + \gamma(\hat{z})\}] \tag{6}$$

The proof of Theorem 1 is composed from three steps:

1. We prove that $G(v, \epsilon)$ is a smooth function of $v, \epsilon$. Therefore, the Hessian of $G(v, \epsilon)$ exists and it is symmetric, namely

$$\partial_v \partial_\epsilon G(v, \epsilon) = \partial_\epsilon \partial_v G(v, \epsilon). \tag{7}$$

2. We show that encoder gradient is apparent in the Hessian:

$$\partial_v \partial_\epsilon G(v, 0) = \nabla_v E_\gamma[\theta(x, z^{\phi_v + \gamma})]. \tag{8}$$

3. We derive our update rule as the complement representation of the Hessian:

$$\partial_\epsilon \partial_v G(v, 0) = \lim_{\epsilon \to 0} \frac{1}{\epsilon}\Big(E_\gamma[\nabla_v \phi_v(x, z^{\epsilon\theta + \phi_v + \gamma}) - \nabla_v \phi_v(x, z^{\phi_v + \gamma})]\Big) \tag{9}$$

The reparameterized gradient computation appears in Equation (10) and it is formally derived in Theorem 1.

**Theorem 1.** *Assume $\phi_v(x, z)$ is a smooth function of $v$. Then*

$$\nabla_v E_\gamma[\theta(x, z^{\phi_v + \gamma})] = \lim_{\epsilon \to 0} \frac{1}{\epsilon}\Big(E_\gamma[\nabla_v \phi_v(x, z^{\epsilon\theta + \phi_v + \gamma}) - \nabla_v \phi_v(x, z^{\phi_v + \gamma})]\Big) \tag{10}$$

*Proof.* First, we prove that $G(v, \epsilon)$ is a smooth function. Recall, $g(\gamma)$ is the zero mean Gumbel probability density function. Applying a change of variable $\hat{\gamma}(z) = \epsilon\theta(x, \hat{z}) + \phi_v(x, \hat{z}) + \gamma(\hat{z})$, we obtain

$$G(v, \epsilon) = \int_{-\infty}^{\infty} g(\gamma) \max_{\hat{z}}\{\epsilon\theta(x, \hat{z}) + \phi_v(x, \hat{z}) + \gamma(\hat{z})\}d\gamma = \int_{-\infty}^{\infty} g(\hat{\gamma} - \epsilon\theta - \phi_v) \max_{\hat{z}}\{\hat{\gamma}(\hat{z})\}d\hat{\gamma}.$$

Since $g(\hat{\gamma} - \epsilon\theta - \phi_v)$ is a smooth function of $\epsilon$ and $\phi_v(x, z)$ and $\phi_v(x, z)$ is a smooth function of $v$, we conclude that $G(v, \epsilon)$ is a smooth function of $v, \epsilon$. Therefore, the Hessian of $G(v, \epsilon)$ exists and symmetric, i.e., $\partial_v \partial_\epsilon G(v, \epsilon) = \partial_\epsilon \partial_v G(v, \epsilon)$. We thus proved Equation (7).

To prove Equations (8) and (9) we differentiate under the integral, both with respect to $\epsilon$ and with respect to $v$. We are able to differentiate under the integral, since $g(\hat{\gamma} - \epsilon\theta - \phi_v)$ is a smooth function of $\epsilon$ and $v$ and its gradient is bounded by an integrable function (cf. Folland (1999), Theorem 2.27).

We turn to prove Equation (8). We begin by noting that $\max_{\hat{z}}\{\epsilon\theta(x, \hat{z}) + \phi_v(x, \hat{z}) + \gamma(\hat{z})\}$ is a maximum over linear function of $\epsilon$, thus by Danskin Theorem (cf. Bertsekas et al. (2003), Proposition 4.5.1) holds $\partial_\epsilon(\max_{\hat{z}}\{\epsilon\theta(x, \hat{z}) + \phi_v(x, \hat{z}) + \gamma(\hat{z})\}) = \theta(x, z^{\epsilon\theta + \phi_v + \gamma})$. By differentiating under the integral, $\partial_\epsilon G(v, \epsilon) = \mathbb{E}_\gamma[\partial_\epsilon(\theta(x, z^{\epsilon\theta + \phi_v + \gamma}) + \phi_v(x, z^{\epsilon\theta + \phi_v + \gamma}) + \gamma(z^{\epsilon\theta + \phi_v + \gamma}))] = \mathbb{E}_\gamma[\theta(x, z^{\epsilon\theta + \phi_v + \gamma})]$. We obtain Equation (8) by differentiating under the integral, now with respect to $v$, and setting $\epsilon = 0$.

Finally, we turn to prove Equation (9). By differentiating under the integral $\partial_v G(v, \epsilon) = \mathbb{E}_\gamma[\nabla_v \phi_v(x, z^{\epsilon\theta + \phi_v + \gamma})]$. Equation (9) is attained by taking the derivative with respect to $\epsilon = 0$ on both sides.

The theorem follows by combining Equation (7) when $\epsilon = 0$, i.e., $\partial_v \partial_\epsilon G(v, 0) = \partial_\epsilon \partial_v G(v, 0)$ with the equalities in Equations (8) and (9). □

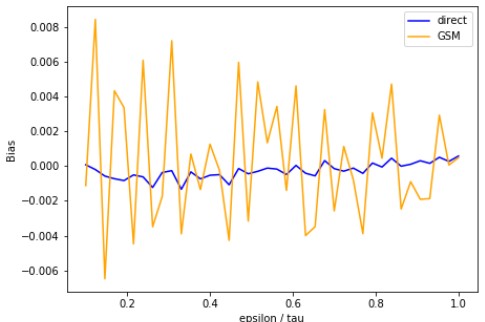 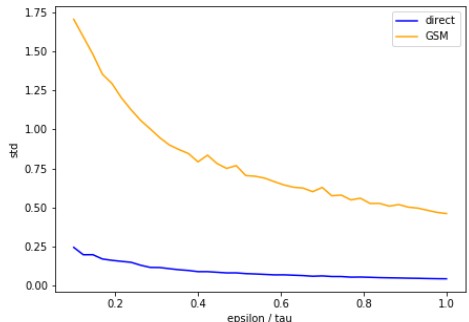

Figure 1: The bias/variance tradeoff of our gradient estimate as a function of $\epsilon$, comparing to Gumbel-Softmax gradient estimate as a function of its temperature $\tau$. The parameters were learned using REINFORCE and from its optimal parameters we estimate the gradient randomly for $500$ times. Left: the average difference from REINFORCE gradient. Right: the average standard deviation of the gradient estimate.

The gradient estimate in Theorem 1 is unbiased in the limit $\epsilon \to 0$. However, for small epsilon the gradient is either zero, when $z^{\epsilon\theta+\phi+\gamma} = z^{\phi+\gamma}$, or very large, since the gradients difference is multiplied by $1/\epsilon$. For large $\epsilon$ we often obtain a moderate non-zero gradient. In practice we use $\epsilon \geq 0.1$ which means that the gradient estimate is biased. This bias-variance tradeoff is evaluated in Figure 1, on an encoder with three layers of sizes $(784, 300, 10)$, a discrete latent structure of size 10 and a matching decoder. In this evaluation we sampled $200$ gradients for a fixed $\epsilon$ to evaluate the bias and standard deviation for each coordinate of the gradient. We report in the graph these quantities.

The above theorem closely relates to the direct loss minimization technique (cf. McAllester et al. (2010); Song et al. (2016)), which, in our setting, can be used to compute the gradient of $\mathbb{E}_x \theta(x, z^\phi)$. The direct loss minimization predicts a single $z^\phi$ for a given $x$ and, therefore, cannot generate a posterior distribution on all $z = 1, ..., k$, i.e., it lacks a generative model that exists in Gumbel-Max perturbation models.

## 5 EXTENSIONS

### 5.1 SEMI SUPERVISED GENERATIVE MODELS

The main advantage in our framework is that learning a discrete VAE using Gumbel-Max reparameterization is intimately related to predicting a discrete latent label through the $\arg\max$ operation $z^{\phi+\gamma}$. Therefore, semi-supervised VAEs are naturally integrated into our reparameterization framework using any loss function. Formally, assume that a subset of the data is labeled, i.e., $S_1 = \{(x_1, z_1), ..., (x_{m_1}, z_{m_1})\}$. In semi-supervised learning, we may add to the learning objective the loss function $\ell(z, z^{\phi+\gamma})$, for any $(x, z) \in S_1$, to better control the prediction of the latent space. The semi-supervised discrete VAEs objective function is

$$\sum_{x \in S} \mathbb{E}_\gamma[\theta(x, z^{\phi+\gamma})] + \sum_{(x,z) \in S_1} \mathbb{E}_\gamma[\ell(z, z^{\phi+\gamma})] + \sum_{x \in S} KL(q_\phi(z|x)||p_\theta(z)) \qquad (11)$$

The supervised component is explicitly handled by Theorem 1 and optimizing a semi-supervised discrete VAEs is straight forward in our framework. Our supervised component is intimately related to direct loss minimization (McAllester et al., 2010; Song et al., 2016), which, in our setting, minimizes $\sum_{(x,z) \in S_1} \ell(z, z^\phi)$. Compared to direct loss minimization, our work adds random perturbation $\gamma$ to the encoder and thus overcomes the "general position" assumption of direct loss minimization. This addition allows us to introduce the "prediction generating function", which greatly simplifies our proof. In addition, the added random perturbation $\gamma$ allows us to use a generative model to prediction, namely, we can randomly generate different explanations $z^{\phi+\gamma}$ while the direct loss minimization allows a single explanation in the form of $z^\phi$.

| $n \times k$ | direct | GSM | REBAR | RELAX |
|---|---|---|---|---|
| $20 \times 10$ | 103.62 | 105.36 | - | - |
| $20 \times 2$ | 126.37 | 128.92 | 127.07 | 126.66 |
| $1 \times 40$ | 145.64 | 145.67 | - | - |

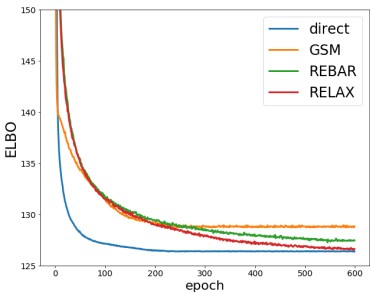

Figure 2: Right: Test log-loss bound of VAEs with different categorial variables $z = (z_1, ..., z_n)$ with $z \in \{1, ..., k\}$. Left: Test log-loss bound of discrete VAE with 20 binary units as a function of training epochs.

Semi-supervised generative models require the discrete space to have a semantic meaning. For example, in generating images of digits, the image may be represented both by the discrete digit class (e.g., $0, 1, ...$) and the continuous style (e.g., bold, tilted,...). Following (Kingma et al., 2014; Jang et al., 2016) we use a mixture of discrete and continuous latent variables in our semi-supervised setting, to capture both the discrete class information and the continuous class style. The network architecture consists of an initial encoder $\phi$ to process the input, which is fed into two separate encoders $\phi_d$ and $\phi_c$ that encode the discrete and continuous latent spaces respectively. The mixed discrete-continuous latent space consists of the matrix $diag(z^{\phi_d + \gamma}) \cdot z_c$, i.e, if $z^{\phi_d + \gamma} = i$ then this matrix is all zero, except for the $i$-th row, which consists of the independent Gaussian random variables $z_c$. An illustration of our mixed discrete-continuous VAE architecture appears in Figure 1. The gradient of the continuous component of the encoder is computed by reparameterization trick. The gradient of the discrete component is computed according to Theorem 1.

## 5.2 DISCRETE PRODUCT SPACES

Discrete VAEs are being applied also to discrete product spaces (Maddison et al., 2016; Jang et al., 2016). A single discrete random variable $z \in \{1, .., k\}$ may not represent the variability of the generative process. Therefore, one may represent the latent space with a discrete product space $z = (z_1, ..., z_n)$, where $z_i \in \{1, ..., k\}$. We note that Theorem 1 holds without any change, if we match the structures of $\theta(x, z)$, $\phi(x, z)$ and $\gamma(z)$, i.e., we have an independent Gumbel random variable for each $z = (z_1, ..., z_n)$. However, this may be computationally inefficient, since the number of Gumbel random variables in this case is exponential in $n$. The gradient in Theorem 1 requires two arg max-perturbations: $z^{\phi+\gamma}$ that requires the encoder output and $z^{\epsilon\theta+\phi+\gamma}$ that requires both the encoder and the decoder log probability. Similarly to Maddison et al. (2016) and Jang et al. (2016) we use a conditionally independent encoder, i.e., $\phi(x, z) = \sum_{i=1}^{n} \phi_i(x, z_i)$ for which we can compute $z^{\phi+\gamma}$ efficiently. In contrast to their approach, we use an approximation for the decoder log probability in order approximate $z^{\epsilon\theta+\phi+\gamma}$ efficiently.

In discrete product spaces we use low-dimensional random perturbation $\gamma(z) = \sum_{i=1}^{n} \gamma_i(z_i)$. Thus instead of exponential number of random variables we use a linear number of random variables (linear in $k$ and $n$). To compute $z^{\phi+\gamma}$ efficiently, we first note that the $\phi(x, z)$ decomposes according to its dimensions, i.e., $\phi(x, z) = \sum_{i=1}^{n} \phi_i(x, z_i)$. Thus the perturb-max argument $z^{\phi+\gamma} = (z_1^{\phi+\gamma}, ..., z_n^{\phi+\gamma})$ also decomposes, i.e., $z_i^{\phi+\gamma} = \arg\max_{\hat{z}=1,...,k} \{\phi_i(x, \hat{z}) + \gamma_i(\hat{z})\}$. Therefore, our approach is exact for the arg max-perturbation $z^{\phi+\gamma}$

Our approximation is not necessarily exact for the arg max-perturbation $z^{\epsilon\theta+\phi+\gamma}$ since the decoder does not decompose according to its dimensions. To be able to compute $z^{\epsilon\theta+\phi+\gamma}$ efficiently, we use the fact we can compute $z^{\phi+\gamma}$ efficiently and apply a low dimensional approximation for the decoder log probability $\tilde{\theta}(x, z) = \sum_{i=1}^{n} \tilde{\theta}_i(z_i)$, where $\tilde{\theta}_i(z_i) = \theta(z^{\gamma_1+\phi_1}, ..., z_i, ..., z^{\gamma_n+\phi_n})$. With this in mind, we approximate $z^{\epsilon\theta+\phi+\gamma}$ using $z^{\epsilon\tilde{\theta}+\phi+\gamma}$, and its coordinates are computed efficiently by $z_i^{\epsilon\tilde{\theta}+\phi+\gamma} = \arg\max_{\hat{z}=1,...,k} \{\epsilon\tilde{\theta}_i(x, \hat{z}_i) + \phi_i(x, \hat{z}) + \gamma_i(\hat{z})\}$.

|  | accuracy | | bound | |
| --- | --- | --- | --- | --- |
| #labels | direct | GSM | direct | GSM |
| 50 | 95.51% | 91.78% | 104.15 | 104.75 |
| 100 | 95.71% | 94.79% | 99.88 | 100.17 |
| 300 | 95.94% | 95.04% | 99.89 | 100.08 |
| 600 | 96.31% | 95.53% | 100.21 | 100.38 |

Figure 3: semi-supervised VAE on MNIST with $50/100/300/600$ labeled examples out of the $50,000$ training examples. Direct loss minimization combined with VAE improves the performance even with weak supervision, e.g., with only 50 examples direct loss minimization with VAE achieves better accuracy than semi-supervised Gumbel-Softmax VAE.

Lastly, the KL-divergence in Equation (1) can utilize the decomposed encoder $\phi(x, z) = \sum_{i=1}^{n} \phi_i(x, z_i)$. In this case $q(z|x) \propto e^{\phi(x,z)}$ and $q(z_i|x) \propto e^{\phi_i(x,z_i)}$, therefore $q(z|x) = \prod_{i=1}^{n} q(z|x)$ and $KL(q_\phi(z|x)||p_\theta(z)) = \sum_{i=1}^{n} KL(q_{\phi_i}(z_i|x)||p_\theta(z_i))$, where $p_\theta(z_i)$ is the marginal probability of $p_\theta(z)$ with respect to its $i-$th entry.

## 6 EXPERIMENTS

We start by comparing our method to the state-of-the-art (Maddison et al., 2016; Jang et al., 2016; Tucker et al., 2017; Grathwohl et al., 2017). We also investigate our approximation for discrete product spaces. We conclude with a set of experiments that demonstrate the effectiveness of our approach in semi-supervised learning of VAEs and the importance of weak supervision in image generation.

### 6.1 DISCRETE VAES

We begin our experiments by comparing the test loss bound of our direct optimization biased gradient estimator with the biased gradient estimator Gumbel-Softmax (GSM) of Maddison et al. (2016) and Jang et al. (2016), and the unbiased estimators REBAR (Tucker et al., 2017) and RELAX (Grathwohl et al., 2017). We performed these experiments using the binarized MNIST dataset (Salakhutdinov & Murray, 2008), and the standard 50,000/10,000 split into training/testing sets. Following Jang et al. (2016) we set our learning rate to $1e - 3$ and the annealing rate to $1e - 5$ and we used their annealing schedule every 1000 step, setting the minimal $\epsilon$ to be 0.1. When considering a latent space with $n$ random variables $z = (z_1, ..., z_n)$ where $z_i \in \{1, ..., k\}$, our encoder has a $(784, n \times k)$ linear layer and the decoder has a $(n \times k, 784)$ linear layer. We applied REBAR and RELAX only to discrete binary random variables[2]. The results appear in Figure 2. For the $20 \times 2$ and $20 \times 10$ architectures, we applied our approximation that is described in Section 5.2.

Next, we explore our discrete product space approximation. We compare a latent space with $n$ random variables $z = (z_1, ..., z_n)$ with $z_i \in \{1, ..., k\}$ with a latent space that has a single random variable $z \in \{1, ..., k^n\}$. When $n = 3, k = 2$, a discrete VAE with a single categorial random variable that has $2^3$ possible values achieved a test log-loss bound of 168.57, while 3 binary variables achieved test log-loss bound of 177.9. When $n = 6, k = 2$, a single categorial random variable achieved a test log-loss bound of 147.59, while 6 binary variables achieved test log-loss bound of 161.09. One can see that there is a gap when using our approximation that increases with $n$.

The main advantage of our framework is that it seamlessly integrates semi-supervised learning. For these experiments, we used a mixed continuous discrete architecture, where the architecture of the encoder consists of a network $\phi$ which has a $(784, 400)$ linear layer, followed by a ReLU, and a $(400, 200)$ linear layer. The output of this later is fed both to a discrete encoder $\phi_d$ and a continuous encoder $\phi_c$. The discrete latent space consists of $z_d \in \{1, ..., 10\}$ and its encoder $\phi_d$ consists of a $(200, 100)$ linear layer, a ReLU, and a $(100, 10)$ linear layer. The continuous latent space considers $k = 10, c = 20$, and its encoder $\phi_c$ consists of a $(200, 100)$ linear layer, a $(100, 66)$ linear layer followed by ReLU and dropout and a $(66, 40)$ linear layer to estimate the mean and variance of $c-$dimensional Gaussian random variables $z_1, ..., z_k$.

---

[2]For REBAR and RELAX we used the code in `https://github.com/duvenaud/relax`.

| w/o glasses | | | | glasses | | | |
| --- | --- | --- | --- | --- | --- | --- | --- |
| woman | | man | | woman | | man | |
| w/o smile | smile | w/o smile | smile | w/o smile | smile | w/o smile | smile |

Figure 4: Learning attribute representation in CelebA, using our semi-supervised setting, by calibrating our $\arg\max$ prediction using a loss function. These images here are generated while setting their attributes to get the desired image. The $i-$th row consists the generation of the same continuous latent variable for all the attributes

Following Kingma et al. (2014), we conducted a quantitive experiment with weak supervision on MNIST with $50/100/300/600$ labeled examples out of the $50,000$ training examples. For labeled examples, we set the perturbed label $z^{\epsilon\theta+\phi+\gamma+\ell}$ to be the true label. This is equivalent to using the indicator function over the space of correct predictions. A comparison of our method with Gumbel-Softmax appears in Figure 3. We can see that direct loss minimization combined with VAE improves the performance even with weak supervision, e.g., with only $50$ examples direct loss minimization with VAE achieves better accuracy than semi-supervised Gumbel-Softmax VAE. We note that we cannot compare the objective function of both methods, as our objective considers direct loss minimization, however, we are able to compare the test log-loss bound, see Figure 3.

Supervision in generative models also greatly helps to control discrete semantics within images. We learn to generate images using $k = 8$ discrete attributes of the CelebA dataset (cf. Liu et al. (2015)) while using our semi-supervised VAE. For this task, we use convolutional layers for both the encoder and the decoder, except the last two layers of the continuous latent model which are linear layers that share parameters over the $8$ possible representations of the image. In Figure 4, we show generated images with discrete semantics turned on/off (with/without glasses, with/without smile, woman/man).

# 7 DISCUSSION AND FUTURE WORK

In this work, we use the Gumbel-Max trick to reparameterize discrete VAEs using the $\arg\max$ prediction operator and show how to propagate gradients through the non-differentiable $\arg\max$ function. We show that this approach compares favorably to state-of-the-art methods, and extend it to semi-supervised learning and image attribute generation.

These results can be taken in a number of different directions. Our gradient estimation is practically biased, while REINFORCE is an unbiased estimator. Our methods may benefit from the REBAR/RELAX framework, which directs biased gradients towards the unbiased gradient (Tucker et al., 2017). There are also open problems when fitting this approach to structured latent spaces (Mena et al., 2018; Jin et al., 2018), as well as estimating its KL-divergence (Roeder et al., 2017). There are also optimization-related questions that arise from our work: the interplay of $\epsilon$ and the learning rate is unexplored and might be correlated. The number of stochastic gradient steps, interleaving Gumbel perturbation with batch samples, might also benefit from a rigorous investigation.

The direct optimization approach we present is general and may be applied beyond VAEs, including reinforcement learning and attention models. Further investigation in this direction is required.

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

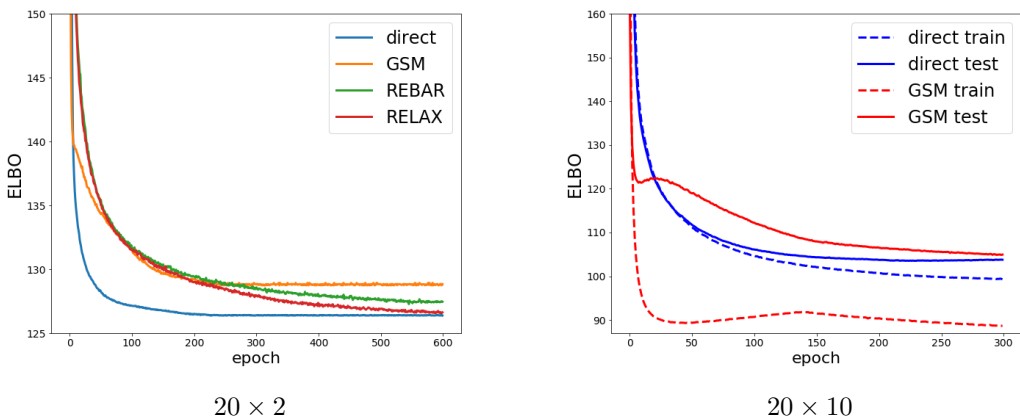

$$20 \times 2 \qquad\qquad 20 \times 10$$

Figure 5: Test log-loss bound of discrete VAE with $n \times k$ binary units as a function of training epochs.

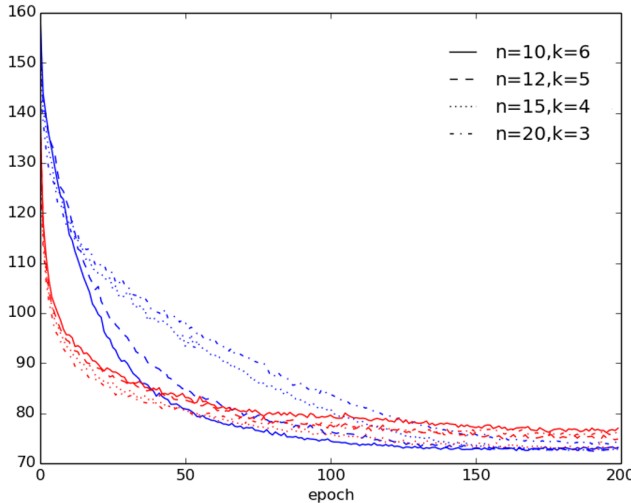

Figure 6: comparing direct and GSM for deep decoder and encoder over discrete product spaces $z = (z_1, ..., z_n)$ where each $z_i \in \{1, ..., k\}$ and $nk = 60$. The encoder is built over a $28 \times 28$ image and has four layers (including input) of $(784, 400, 200, 60)$ units respectively. The decoder architecture has four layers with $(60, 200, 400, 784)$ units respectively, including the $28 \times 28$ output layer. Between each layer we have a ReLU function and the output layer used $\tanh$ to calibrate the pixel value. The experiment is performed over non-binarized MNIST with the MSE loss.

## A  GUMBEL-MAX PERTURBATION MODEL AND THE GIBBS DISTRIBUTION

**Theorem 2.** *Gumbel & Lieblein (1954); Luce (1959); McFadden (1973) Let $\gamma$ be a random function that associates random variable $\gamma(z)$ for each $z = 1, ..., k$ whose distribution follows the zero mean Gumbel distribution law, i.e., its probability density function is $g(t) = e^{-(t+c+e^{-(t+c)})}$ for the Euler constant $c \approx 0.57$. Then*

$$\frac{e^{\phi(x,z)}}{\sum_{\hat{z}} e^{\phi(x,\hat{z})}} = \mathbb{P}_{\gamma \sim g}[z = z^{\phi+\gamma}], \text{ where } z^{\phi+\gamma} \stackrel{def}{=} \arg\max_{\hat{z}=1,...,k} \left\{ \phi(x,\hat{z}) + \gamma(\hat{z}) \right\} \tag{12}$$

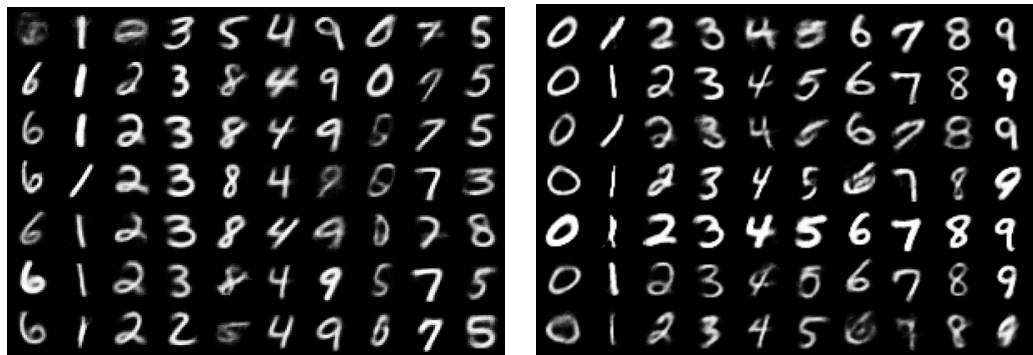

Figure 7: Comparing unsupervised to semi-supervised VAE on MNIST, for which the discrete latent variable has 10 values, i.e., $z \in \{1, ..., 10\}$. One can see that semi-supervised helps the VAE to capture the class information and consequently improve the image generation process. Left: generated images from an unsupervised VAE. Right: generated images from semi-supervised mixture model. The $j-$th column consists of images generated for the $j-$th discrete class. The $i-$th row consists the generation of the same continuous latent variable for all the 10 classes. One can see that the discrete latent space is able to learn the class representation. The continuous latent space learns the variability within the class. Comparing to unsupervised image generation, we observe that some supervision improves the image generation per class.

*Proof.* Let $G(t) = e^{-e^{-(t+c)}}$ be the Gumbel cumulative distribution function. Then

$$\mathbb{P}_{\gamma \sim g}[z = z^{\phi+\gamma}] = \mathbb{P}_{\gamma \sim g}[z \in \arg \max_{\hat{z}=1,..,k} \{\phi(x,\hat{z}) + \gamma(\hat{z})\}] \tag{13}$$

$$= \int g(t - \phi(x,z)) \prod_{\hat{z} \neq z} G(t - \phi(x,\hat{z}))dt \tag{14}$$

Since $g(t) = e^{-(t+c)}G(t)$ it holds that

$$\int g(t - \phi(z)) \prod_{\hat{z} \neq z} G(t - \phi(\hat{z}))dt = \int e^{-(t-\phi(x,z)+c)}G(t - \phi(x,z)) \prod_{\hat{z} \neq z} G(t - \phi(x,\hat{z}))dt$$

$$= \frac{e^{\phi(x,z)}}{Z} \tag{15}$$

where $\frac{1}{Z} = \int e^{-(t+c)} \prod_{\hat{z}=1}^{k} G(t - \phi(\hat{z}))dt$ is independent of $z$. Since $\mathbb{P}_{\gamma \sim g}[z = z^{\phi+\gamma}]$ is a distribution then $Z$ must equal to $\sum_{\hat{z}=1}^{k} e^{\phi(x,\hat{z})}$. $\qquad \square$

| w/o glasses | | | | glasses | | | |
| --- | --- | --- | --- | --- | --- | --- | --- |
| woman | | man | | woman | | man | |
| w/o smile | smile | w/o smile | smile | w/o smile | smile | w/o smile | smile |

Figure 8: Extending Figure 4. The $i-$th row consists the generation of the same continuous latent variable for all the attributes.

