# OpenReview forum: "Direct Optimization through $\arg \max$  for  Discrete Variational Auto-Encoder"
_ICLR.cc/2019/Conference_

### Official Review · AnonReviewer1 · 2018-10-30
**An principled approach with weak empirical results**

**Rating:** 5
**Confidence:** 4

**Review:**

This work proposes a new (biased) gradient estimator to learn a discrete auto-encoders. Similarly to the gumbel-softmax estimator this paper proposes to use the gumbel-max trick and the reparametrization trick but instead of relaxing the argmax by a softmax, the authors derive a formula for the gradient based on direct loss optimization to compute the gradient through the argmax.

Pros:
- The approach is well motivated and the proof of theorem 1 which gives the formula of the new gradient estimator seems correct.

Cons:
- The principle downside of the proposed approach is that it requires to compute the value of the objective for several values of z, which makes it more computationally expensive than gumbel-softmax. Could the author compare the different estimators in terms of running time instead of epoch for example in fig2. it seems like RELAX would perform similarly or better in terms of wall-clock time.

- [1] also proposed an estimator that requires evaluating the objective for different values of z, and showed that it is unbiased and optimal (lowest variance). I think the authors should mention this related work and how their approach differs. I also think the author should compare their work to [1].

- Since both gumbel-softmax and the proposed approach are biased, could the authors give some intuitions on why they believe their approach is better.

- I believe the expectation of the right-hand side of equation (9) can be computed in closed form by using a formula similar to eq (4) and (5), which replace the expectation by a sum over the possible values of z. This will lead to a gradient estimator with no variance, can the author comment on this ?

- I think the bias induced by the mean-field approximation of the decoder should be investigated more thoroughly. Could the authors plot the gap as a function of n for example ? What happens if we also increase the number of category ? (there is a typo in this section it should be k^n instead of n^k) ? Can they compare to gumbel-softmax, is there a threshold at which gumbel-softmax becomes better ?

- It's not clear on what setting is the variance plotted in fig 1. is computed ? Is it computed on the discrete VAE experiment ? if so how many latent variables and category ? Could the bias also be provided ? Could it be compared to gumbel-softmax with varying temperature ?

- The experiments are a bit toyish, it's not clear what happens when the task are more complex, the architecture for the encoder and decoder are deeper or the latent space is bigger. In particular the authors only consider linear encoder and decoder when comparing the ELBO of different methods.

- In the semi-supervised settings what happens if we don't set the perturbed level to the true label ?

Conclusion:
The experiments are quite toyish and the approach is more computationally expensive than gumbel-softmax. More experiments should be done to clearly show the advantage of this method compared to gumbel-softmax.

---

> ### Author Response · Authors · 2018-11-16
> **Thank you for your constructive comments**
>
> We thank you for your thoughtful comments. Here are answers to the concerns raised by the review. We complement these answers with a revised submission (appendix and paper)
>
> Wall-clock time for the CelebA with $10$ binary attributes takes 0.13 seconds for Gumbel-Softmax and 0.06 seconds for Gumbel-Max, when the discrete latent space is structured (spin glass model). More generally: Gumbel-Softmax uses in its forward computation: (i) one computation of $\theta$ (getting as input a softmax) and (ii) $|{\cal Z}|$ computations of $\phi$ for each $z \in {\cal Z}$ (due to the normalization of the softmax). In its backward computation it uses: (i) one gradient for $\theta$ and (ii) $|{\cal Z}|$ computations derivatives of $\phi$ (due to the normalization of the softmax), for each $z \in {\cal Z}$. In contrast, Gumbel-Max uses in its forward computation (i) one computation of $\theta$ (getting as input the argmax) and (ii) max computation over $\phi$. In its backward computation it requires another max operation (total of two max operations), now with $\theta$ (when the $\theta$ is decomposable, as in our discrete product space approximation, this is as efficient as computing the maximum over $\phi$) and two gradient computations of $\phi$.
>
> The computational complexity of the max operation is sometimes much less than the computational complexity of the normalization constant since the max operation does not necessarily requires going over all $z \in {\cal Z}$ in all cases. This happens in structured prediction models, where the max operation can be computed with integer linear solvers efficiently. Structured prediction models are important in practice since they capture correlations between labels (as in the CelebA problem).
>
> The work [1] in its one dimensional form consdiers $\sum_z e^{\phi_v(x,z)} \theta(x,z)$ (assuming the normalization constant is one for simplicity). Comparing to Gumbel-Softmax and Gumbel-Max, it requires in its forward computation $|{\cal Z}|$ computations of $\phi$ and $\theta$ and in its backward computation $|{\cal Z}|$ computations of $\phi$ and $\theta$ gradients. We agree that such a comparison is in place, we will compare the works.
>
> We agree that Eq 9 can be computed similarly to Eq 4, Eq 5, it will result in REINFORCE for Gumbel-Max: The update rule of variational Bayes in its discrete setting is $\sum_z e^{\phi_v(x,z)} \theta(x,z) \nabla_v \phi_v(x,z)$ (assuming the normalization constant is one for simplicity). When doing it in the perturbation space, this will lead to REINFORCE update rule with respect to perturbation models, and the gradient is  $ \mathbb{E}_{\gamma \sim g} [ \nabla \log(g(\gamma)) \nabla \phi(x,z^{\phi + \gamma }) \theta(x,z^{\phi + \gamma})] $ where $g$ is the probability density function of the perturbation. This representation perhaps gives some insight for the variance of REINFORCE (multiplying the gradient by a log-gradient and $\theta$, compared to our (biased) approach.
>
> We added a plot (Figure 6 in the Appendix), also experimenting with depth and different loss functions. When $n$ increases the gap becomes smaller and the difference is negligible (and random).
>
> The variance experiment was conducted as follows: the network was trained over all the samples until convergence, then computed the encoder gradients 1000 times (with 1000 different Gumbels)  over one sample and computed the variance over the 1000 gumbels.  We will add a comparison to Gumbel-Softmax with temperature and the bias as well. however, we are not sure that $\epsilon$ is the right equivalent to temperature in Gumbel-Softmax. Instead, we think that the analogue is the perturbation variance: the temperature signifies the max-argument, as happens when the variance of the perturbation approach zero. $\epsilon$ in our setting inserts the gradient signal
>
> In our experiments, we focused on the settings of Gumbel-Softmax, to be able to compare to the previous methods. For structured setting we worked on CelebA but limited ourselves to small number of discrete variables to be able to compare to Gumbel-Softmax. In retrospect, we should have emphasized the structured encoder setting, where our method excels (when we compute two max operations instead of summing over all structures), and we will elaborate on that.
>
> In semi-supervised setting, we can plug any loss. In the MNIST task it sometimes perform better and sometimes worse. Our setting was chosen from computational aspect, since it allowed us to set the perturbed prediction to the true label.

---

### Official Review · AnonReviewer2 · 2018-11-04
**A significant contribution**

**Rating:** 7
**Confidence:** 4

**Review:**

The authors propose a method to apply the reparametrization trick when the random variables of interest are discrete. Their technique is based on a formulation of the objective function in terms of Gumbel-Max operators. They propose a derivation of the gradient in terms of an auxiliary variable \epsilon, such that the resulting gradient estimate is biased but the bias is reduced as \epsilon approaches zero, at the cost of increasing variance. Experiments are performed with VAE including discrete latent variable models. The authors show how their method converges faster than other baselines formed by estimators of the gradient given by the REBAR, RELAX and Gumbel-soft-max methods. In experiments with semi-supervised VAEs, their method outperforms the Gumbel softmax method in terms of accuracy and objective function.

Quality:

The theoretical derivations seem rigorous and the experiments performed clearly indicate that the proposed method can outperform existing baselines.

Clarity:

The paper is clearly written and easy to read. I found that the network architecture shown in the left of Figure 1 a bit confusing and needs to be explained more clearly.

Significance:

The experimental results clearly show that the proposed method can outperform existing baselines and that the proposed contribution is significant.

Novelty:

The proposed method is novel up to my knowledge. This is the first time I have seen the proposed theoretical derivations, which are significantly different from previous approaches.

---

### Official Review · AnonReviewer3 · 2018-11-10
**Worthwhile and interesting paper, but exposition could use some work (rating maintained after author feedback).**

**Rating:** 7
**Confidence:** 4

**Review:**

This paper proposes combining the Gumbel-max trick and "direct loss optimization" for variance reduction in VAEs with discrete latent variables. This is a natural combination (in hindsight), since the Gumbel-max trick turns sampling into non-differentiable optimization, and direct loss optimization provides a way to optimize the expected value of a non-differentiable loss. The paper is well-written for the most part and is backed by good experimental results. However it like some of the mathematical details and some of the exposition could be greatly improved.

I think there are several mistakes in the reasoning presented in the proof of Theorem 1 (see detailed comments below). Theorem 1 in the current paper seems to me to be a special case of Theorem 1 in (Song 2016), where the expectation over data is replaced by an expectation over the Gumbel variable gamma. If I've understood correctly, it seems like it would be more correct and concise to simply cite that paper with some explanatory comments.

The word "direct" occurs quite a lot in the paper. It sometimes seemed misplaced. For example for "The direct differentiation of the resulting expectation" in the introduction, in what sense is the differentiation direct, and what would non-direct differentiation be?

In section 3, that's not the meaning of the term "exponential family".

The re-use of theta and psi as both model parameters and the log probability density / distribution is unnecessarily confusing.

A small point, but in "the challenge in generative learning is to reparameterize and optimize (2)", the authors assume that q has analytic expression for the second KL term in (1). That's often the case but definitely not always. Also, even if this KL term has an analytic expression, it is not always better to use it (see Duvenaud "Sticking the landing...").

In (3), the usual notation is P(x = i) where x is the random variable and i is its possible value, whereas in (3) the random variable z^{\phi + \gamma} appears on the right of the equals sign.

The first paragraph of section 4.1 and (4) and (5) are just a simple application of the law of total expectation, and it would be simpler and clearer to state that.

"gradient of the decoder" should be "gradient of the decoder log probability" (or log prob density depending on preference). Similarly with "the decoder is a smooth function". The decoder is a conditional probability distribution (at least according to my understanding of conventional usage).

In the first equation in the proof of Theorem 1, it seems as though the authors are using the standard change of variables formula for integrals. However the new variable \hat{\gamma} depends on \hat{z} through \theta, so I don't see how it's valid to ignore the max in the way the present paper does. One way to see that something is wrong is the fact that the integrand on LHS has \hat{z} as a bound variable only, whereas the integrand on RHS has \hat{z} as both a bound variable (inside the max) and a free variable (since \theta depends on \hat{z}, though strangely that is not written in the equation). What is the value of \hat{z} used for \theta on RHS?

There's a missing [] after \partial_\epsilon in the third line of the paragraph starting "We turn to prove Equation (8)".

In the same line, I don't see why the two expectations are equal. It seems to me that the differentiation w.r.t. epsilon ignores the fact that changing epsilon occasionally changes z^{\epsilon \theta + \phi_v + \gamma} in a discontinuous way. The term being differentiated has both a continuous-in-epsilon component and a piecewise-constant-in-epsilon component, and the latter appears to have been ignored. While the gradient of a piecewise constant function is zero almost everywhere, the occasional large changes (which could be thought of as delta functions) still can make a large contribution to the overall expression once we take the expectation. To look at it another way, if the reasoning here is correct, why can't the same argument be used on the RHS of (8), first to take the derivative inside the expectation and subsequently to compute the derivative as zero, since the inner term is a piecewise constant function of v? Yet clearly the RHS of (8) is not always zero.

Around "However when we approach the limit, the variance of the estimate increases...", I think it would be extremely helpful to explain that for small epsilon, we occasionally obtain a large gradient (and otherwise zero), while for large epsilon we often obtain a moderate non-zero gradient. That gives some insight into the effect of epsilon, and why the variance is larger for small epsilon.

Any reason not to plot the bias in right Figure 1, which is ostensibly about the bias-variance trade-off?

I didn't follow the meaning of the diagram or caption for left Figure 1.

In (11), I wasn't sure whether S included the supervised examples or not (i.e. whether S_1 was disjoint from or a subset of S). If disjoint, shouldn't the KL term be included, or the expectation-over-gamma term be changed to use ground truth z? I guess I was unclear on the form of loss used for the supervised data, and unclear on the motivation for this choice.

In the last sentence of section 5.1, should "chain rule" be "variance reparameterization trick"?

In section 5.2, it would be helpful to mention what mean field means in terms of the variational distribution q (namely q(z | x) = \prod_i q(z_i | x) ). Also, the term "mean field" is not conventionally used for general distributions (such as the decoder here) as far as I'm aware, only for variational distributions. "Conditionally independent" might be clearer.

What does "for which we can approximate z^{...} efficiently" refer to?

In section 6.1, what is the annealing rate? Also, the minimal epsilon is set to 0.1. Is epsilon changed as training progresses according to some schedule?

"The main advantage of our framework is that it seamlessly integrates semi-supervised learning" seems like an overstatement. Wouldn't semi-supervised learning be relatively straightforward to incorporate into any form of VAE? And why not just use log p(x, z) for updating the decoder parameters and log q(z | x) for updating the encoder parameters?

How many labeled examples were used for the CelebA semi-supervised learning?

Some bibliography typos. For example, no capitalization throughout (e.g. "gumbel" instead of "Gumbel"). Also lots of arxiv preprints cited when published papers exist (e.g. Jang 2017 should be ICLR 2017 not arxiv preprint).

---

> ### Author Response · Authors · 2018-11-18
> **Thank you for your insightful suggestions**
>
> We thank you for your time and effort. Many of your comments helped us to improve the submission. We uploaded a new manuscript with all suggestions.
>
> We agree that citing Song 2016 will solve these issues, but we think that these issues are a result of poor notation rather than a mistake: we tried to condense Danskin theorem to one line, and implicitly work with product spaces that decouple the dependencies of $\hat z$, it was a mistake as it made our derivation unclear.
>
> The first equation in Theorem 1:
> This is a convolution between a smooth function and a non-smooth function, and therefore it is smooth. Take for example a two dimensional Gaussian random variable with mean $\mu$. The expectation of their max is a smooth function. Analytically, it equals to $C\int_{-\infty}^\infty \int_{-\infty}^\infty  e^{\|\gamma - \mu\|^2/2} \max\{\gamma_1,\gamma_2\} d \gamma_1 d \gamma_2$. While $\max \{\gamma_1,\gamma_2\}$ certainly depends on $\gamma$ through $\mu$, the integral is a smooth function of $\mu$. We defined $g(\gamma)$ more precisely above equation 3 to show it is a product space so it decouples dependencies of $\theta$ within $\hat z$: The notation $g(\hat \gamma - \epsilon \theta - \phi_v )$ implicitly uses the independent product space $\prod_{\hat z} g(\hat \gamma(x,\hat z) - \epsilon \theta(x,\hat z) - \phi_v(x,\hat z) )$. We borrowed this notation from Gaussian random variables, where in this case it is $e^{\|\gamma - \mu\|^2/2}  = \prod_{\hat z} e^{(\gamma(\hat z) - \mu(\hat z))^2/2}$.
>
> "We turn to prove Equation 8" paragraph:
> We agree we chose poor notation. The function $\max_{\hat z} \{\epsilon \theta(x,\hat z) + \phi_v(x,\hat z) + \gamma(\hat z) \}$ is the maximum of linear functions of $\epsilon$. Therefore Danskin Theorem (Proposition 4.5.1 in Convex Analysis and Optimization by Bertsekas) states that $\partial_\epsilon(\max_{\hat z} \{\epsilon \theta(x,\hat z) + \phi_v(x,\hat z) + \gamma(\hat z) \}) = \theta(x,z^{\epsilon \theta + \phi_v + \gamma})$ whenever the $\arg \max$ is unique. Since the $\arg \max$ is unique with probability one, we can continue without problems, thus overcoming the general position condition in McAllester et al 2010 and the regularity conditions in Song et al. 2016.
>
> Clarifications:
> * Annealing: We set the annealing rate to be 1e-3 to follow Jang et al. We stop at $0.1$ to avoid gradient blowup.
>
> * Semi-supervised: we referred to general loss functions, beyond the log-loss. We agree that the log loss is a natural choice but there are recent cases where semi-supervised is important and log-loss cannot capture the structures accurately. Such losses can extend Corro and Titov's "Differentiable Perturb-and-Parse: Semi-Supervised Parsing with a Structured Variational Autoencoder"}
>
> * We used 500 examples in CelebA
>
> * We will fix the biography
>
> * We will add the bias in our bias/variance tradeoff

---

> > ### Comment · AnonReviewer3 · 2018-12-04
> > **Reviewer response**
> >
> > I appreciate the authors' feedback, and much of it was very helpful. However, I don't feel like the authors did a very good job of addressing my comments.
> >
> > Regarding Theorem 1, after carefully reviewing the authors' comments and going through some working of my own, I now believe each of the claims in Theorem 1 is correct, and some of my confusion was indeed solely due to non-standard notation rather than incorrect reasoning. It is quite a nice way to derive (10)! However the presentation still leaves a huge amount to be desired even after the authors' improvements. Firstly, the differentiating-under-the-integral argument justifies differentiating under the integral for the expression on the right side of the first equation in the proof, but not for the expression on the left side, which is what is actually used. Please clarify the reasoning here. Secondly, for Folland (1999), Theorem 2.27, we require df/dt to exist (in their notation), but it does not exist everywhere (only almost everywhere) for the max function according to a strict definition of derivative. Please clarify why this theorem actually applies here. Also, what precisely is the integrable function which bounds the derivative? The Danskin argument is used to justify the derivation of equation (8), but the derivation of equation (9) also requires a similar derivative-of-max argument, and Danskin is not completely straightforward to apply in this case since the part inside the max is not convex.
> >
> > I would suggest that it might be easier to first define the integral of $g(\gamma) max_z{v(z) + \gamma(z)} d\gamma$, derive the derivatives of this with respect to $v$, then use this to derive both (8) and (9).
> >
> > A few other small points in the proof of Theorem 1. It might be worth reiterating that $g$ is the *multivariate* Gumbel pdf to stress that the integral is not 1D. Using the limits -infty and infty also seems strange, and suggests a 1D integral. I would suggest omitting those limits. The notation $\theta$ makes sense now; I had not realized it was intended to be a vector with components indexed by $\hat{z}$. I would suggest using the more standard subscript notation $\theta_{\hat{z}}$ or just $\theta_i$ for vectors. The notation $\theta$ also doesn't work very well as it drops the dependence on $x$ (though I understand this is not relevant for the proof). The change of variables in the first equation in the proof also now makes sense to me. It would be helpful to state the result being used that the convolution of a smooth function and a non-smooth function is smooth, and also state this result precisely; it is not true in general for non-compactably-supported functions if I understand correctly. Incidentally it would also be helpful to precisely define "smooth". The meaning I presume is intended (infinitely continuously differentiable) is fairly standard but other definitions are also used. For "differentiating under the integral, now with respect to v" in the paragraph starting "We turn to prove Equation (8)", there is actually no need to differentiate under the integral to obtain the desired result, which is good because if I understand correctly differentiating under the integral here would not be valid. Also, "taking the derivative with respect to $\epsilon = 0$" is awkward shorthand for "taking the derivative with respect to $\epsilon$ and setting $\epsilon$ to 0"; I would use the more explicit version. Also, in "Applying a change of variable...", there is a missing hat on $z$.
> >
> > I still think it's worth mentioning that the key result (10) can also be derived using Theorem 1 in the Song paper (briefly alluding to any regularity conditions needed). This would help give additional confidence that the result is correct.
> >
> > Finally on Theorem 1, I think it would be helpful for the reader's intuition to state why Folland (1999) Theorem 2.27 does not apply to differentiating under the integral of expressions like the right side of (8). Intuitively to me the difference is that the gradient of the integrand in (8) is a delta function, whereas in (6) the gradient of the integrand was merely discontinuous, but bounded. Some clarifying statement would be helpful for the reader to understand when the sort of reasoning the authors are using is valid or not.

---

> > > ### Comment · AnonReviewer3 · 2018-12-04
> > > **Reviewer response (continued)**
> > >
> > > For the annealing rate, I actually meant what is the annealing rate in the sense of what does it mean as a phrase? Even if defined in Jang et al. it would be useful to briefly restate here.
> > >
> > > Also, I still didn't see any explanation as to whether epsilon changed as training progresses according to some schedule?
> > >
> > > Thanks for your clarification regarding the log loss. "The main advantage of our framework is that it seamlessly integrates semi-supervised learning" is still an overstatement, since many forms of VAE can incorporate semi-supervised learning easily. Also, some of the clarification should go in the paper, not just these review notes.
> > >
> > > The bibliography still contains a ton of miscapitalizations, e.g. "john wiley & sons". Please fix.
> > >
> > > In figure 1, I suggest using more than 500 samples. The GSM plot is so noisy it is hard to compare the two biases, and the trade-off is hard to see at the moment since the bias appears essentially flat for "direct" due to the noisiness and the scale of the y axis.
> > >
> > > For the law of total expectation point, I actually meant that the entire first paragraph of section 4.1 and (4) and (5) are just a simple application of the law of total expectation, and could be replaced by simply stating that and the left side of (4) and the right side of (5).
> > >
> > > There were a number of other comments that did not appear to be addressed:
> > >
> > > The re-use of theta and psi as both model parameters and the log probability density / distribution is unnecessarily confusing.
> > >
> > > A small point, but in "the challenge in generative learning is to reparameterize and optimize (2)", the authors assume that q has analytic expression for the second KL term in (1). That's often the case but definitely not always. Also, even if this KL term has an analytic expression, it is not always better to use it (see Duvenaud "Sticking the landing...").
> > >
> > > In (11), I wasn't sure whether S included the supervised examples or not (i.e. whether S_1 was disjoint from or a subset of S). If disjoint, shouldn't the KL term be included, or the expectation-over-gamma term be changed to use ground truth z? I guess I was unclear on the form of loss used for the supervised data, and unclear on the motivation for this choice.
> > >
> > > The additional sentence after "for which we can approximate z^{...} efficiently" is very opaque and does not really make explicit what the authors are referring to. Explicit equations would be helpful, please!

---

### Meta-Review · Area_Chair1 · 2018-12-18
**Original approach, but with unclear computational benefit**

**Confidence:** 4
**Recommendation:** Reject

**Metareview:**

The paper presents a novel gradient estimator for optimizing VAEs with discrete latents, that is based on using a Direct Loss Minimization approach (as initially developed for structured prediction) on top of the Gumble-max trick. This is an interesting and original alternative to the use of REINFORCE or Gumble Softmax. The approach is mathematically well detailed, but exposition could be easier to follow if it used a more standard notation. After clarifications by the authors, reviewers agreed that the main theorerm is correct. The proposed method is shown empirically to converge faster than Gumbel-softmax, REBAR, and RELAX baselines in number of epochs. However, as questioned by one reviewer, the proposed method appears to require many more forward passes (evaluations) of the decoder for each example. Authors replied by highlighting that an argmax can be more computationally efficient than softmax (in cases when the discrete latent space is structured), and also clarified in the paper their use of an essential computational approximation they make for discrete product spaces. These are important aspects that affect computational complexity. But they do not address the question raised about using significantly more decoder evaluations for each example. A fair comparison for sampling based gradient estimation methods should rest on actual number of decoder evaluations and on resulting timing. The paper currently does not sufficiently discuss the computational complexity of the proposed estimator against alternatives, nor take this essential aspect into account in the empirical comparisons it reports.
We encourage the authors to refocus the paper and fully develop and showcase a use case where the approach could yield a clear a computational advantage, like the structured encoder setting they mentioned in the rebuttal.

---

> ### Public Comment · ~Mingzhang_Yin1 · 2018-12-21
> **Related reference**
>
> I enjoy reading the gradient estimation methods with multiple evaluations in the paper!
> I would like to point our concurrent work which also provides low variance, unbiased gradient estimator for discrete latent variables which may serve as a proper comparison.  https://openreview.net/pdf?id=S1lg0jAcYm